# Comparative Study of Inhaled Fluticasone Versus Oral Prednisone in 30 Dogs with Cough and Tracheal Collapse

**DOI:** 10.3390/vetsci10090548

**Published:** 2023-09-01

**Authors:** Jesús Talavera-López, Oscar Sáez-Mengual, María-Josefa Fernández-del-Palacio

**Affiliations:** 1Cardiorespiratory Service, Veterinary Teaching Hospital of University of Murcia, Espinardo, 30100 Murcia, Spain; mjfp@um.es; 2Veterinary Clinic 7 Vidas, C/Dr Van Der Hofstan, s/n, San Juan, 03550 Alicante, Spain; oscar@veterinaria7vidas.com

**Keywords:** aerosol therapy, metered-dose inhaler (MDI), airway inflammation, valved holding chamber, fluticasone propionate, prednisone

## Abstract

**Simple Summary:**

Coughing is common in dogs with tracheal collapse (TC). Inhaled corticosteroids are the standard of care in humans, but their use in dogs is less widespread than oral corticosteroids. This study compared the efficacy, adherence, and tolerance to treatment of inhaled versus oral corticosteroids in 30 dogs with cough and TC, randomized to each treatment. Responses were monitored in the hospital (enrolment and weeks 2 and 4) and at home (weeks 1 to 4) using a semiquantitative clinical scale based on four key clinical parameters (respiratory distress, cough episodes, cough frequency, tracheal sensitivity). Both treatments provided comparable clinical improvements, with significantly greater improvements at the end in the fluticasone group. Adherence and tolerance were equivalent, but thirst and the frequency of urination increased in those taking prednisone. In conclusion, inhaled fluticasone is effective in controlling cough in dogs with TC without side effects, which encourages its use to be extended compared to prednisone.

**Abstract:**

Coughing is common in dogs with tracheal collapse (TC). The use of inhaled corticosteroids is less widespread than oral ones. This study aims to compare the effects of oral and inhaled corticosteroids in dogs with cough and TC. Thirty dogs were prospectively included and randomized to the prednisone oral group (OG, 14) or fluticasone inhaled group (IG, 16). A clinical score (CS) based on four clinical parameters (respiratory distress, cough episodes, cough frequency, tracheal sensitivity) was monitored at the hospital (enrolment and weeks 2 and 4). Water intake, urination habits, and adherence and tolerance to treatments were monitored weekly. Significant improvements in clinical parameters were identified in both groups throughout the study. Between-group (OG–IG) comparisons revealed no significant differences, indicating equivalent improvement. At the study’s endpoint, the IG dogs had a significantly lower CS (5.69 ± 0.79) than OG dogs (6.43 ± 1.02, *p* < 0.05). Adherence and tolerance were comparable. From weeks 2 to 4, OG dogs were significantly thirstier and urinated more frequently than IG dogs. In conclusion, fluticasone provided good tolerability and efficacy in controlling cough in dogs with TC, and they showed a lower incidence of signs of hypercortisolism compared to prednisone. These data encourage the use of inhaled fluticasone in dogs with cough and TC.

## 1. Introduction

Tracheal collapse (TC) in dogs is a degenerative structural disorder of the tracheal wall, resulting in the excessive (usually dorsoventral) dynamic collapse of the tracheal lumen [1,2] whenever the extraluminal pressure exceeds the intraluminal pressure. The apposition of the tracheal mucosa, as well as the turbulent air through a narrow tracheal lumen, may induce respiratory difficulty, tracheal stridor, and coughing. All of this predisposes to chronic inflammation, inducing a vicious circle [3,4].

The complete etiology has not been established. A multifactorial origin, including congenital anomalies (chondromalacia) and acquired conditions (such as chronic diseases of airways, cartilage degeneration and trauma), seems to be implicated [2,3,5,6,7,8]. Traditionally, TC has been described as an isolated respiratory disorder in dogs, but more recently, an increasing body of research has documented its occurrence with multiple other cardiorespiratory diseases [4]. Structurally, the tracheal rings of affected dogs show deficiencies of glycoproteins, glycosaminoglycans and calcium, as well as decreased water content, resulting in loss of cartilage rigidity [2,6]. There is a greater rate of incidence in small dogs, and it is especially frequent in Yorkshire terriers, Maltese Bichons and Pomeranians [2,3,7]. Occasionally, it has been described in large breeds [9].

In predisposed breeds, the diagnosis of TC is very often presumptively established based on the case history, clinical signs, physical examination and radiography of the neck and thorax. One of the most common signs of TC is a dry, harsh and persistent cough, sometimes described as a “goose-honking” cough, which can progress to a wheezing noise and difficulty breathing. Radiography is a definitive indication in the first-step evaluation of dogs with suspicion of TC, and allows one to discard other possible etiologies and to identify concomitant cardiorespiratory diseases. Complementary techniques such as fluoroscopy, computed tomography, endoscopy, and ultrasonography contribute to the diagnosis and evaluation of the severity of the process [1,2,3,7,10].

The concurrence of coughing and increased sensitivity to palpation of the trachea may indicate the presence of associated tracheal inflammation, whose treatment may provide general improvements in the patient by interrupting the vicious circle of airway collapse–inflammation–cough. In these cases, an indication for the use of corticosteroids may exist. 

In humans and cats with airway disease, inhaled corticosteroids have shown proven efficacy for cough control and are considered the standard of care in place of oral corticosteroids [11,12,13,14,15]. Recent studies have determined that 78% of board-certified specialists use prednisone/prednisolone as the first option in these cases, and only 21% use inhaled fluticasone [4]. Many of them (50%) permit their use for long periods of time. However, chronic oral corticotherapy can induce systemic adverse effects, such as polyphagia, polydipsia–polyuria, myopenia and lethargy, which can complicate the medical management of chronic cough [16]. Also, it can induce another series of problems, such as cartilage degradation [17] and a predisposition for obesity [18], which can, in turn, contribute to the worsening of the clinical signs of the patient. Oral corticosteroids therapy is contraindicated in dogs with renal disease, diabetes mellitus or advanced cardiac disease [19], all of which coexist very frequently in dogs with TC.

The use of inhaled corticosteroids has recently demonstrated its efficacy in the treatment of dogs with TC and other causes of cough [20]. The inhaled treatment makes it possible to directly concentrate the pharmacological benefits to the target tissue and mostly avoid its negative consequences at a systemic level, with slight suppression of the hypothalamic–pituitary–adrenal axis (HPAA) [21,22,23]. This is especially interesting for patients with chronic diseases that require prolonged or intermittent high-frequency regimens.

Despite the theoretical clinical benefits, no prospective studies have compared the therapeutic effects of inhaled versus oral corticosteroids in dogs with cough and TC. The availability of these data is timely and necessary in seeking to improve penetrance among specialists and general practitioners. The goal of this study was to evaluate the comparative efficacy of oral versus inhaled corticosteroids parallel to the appearance of polyuria–polydipsia as markers of iatrogenic hyperadrenocorticism and differences in tolerance and adherence to each treatment regimen over a 4-week period. We hypothesized that inhaled corticosteroids would provide positive responses comparable to oral ones, with less concurrence of polyuria–polydipsia, and good tolerance and adherence to treatment.

## 2. Materials and Methods

Client-owned dogs presented to the Cardiorespiratory Service of the Veterinary Teaching Hospital of the University of Murcia and the Veterinary Hospital “7 Vidas” for evaluation of cough were considered candidates for inclusion in this prospective observational randomized study. Owners provided informed consent, and the Institutional Animal Care and Use Committee at the University of Murcia approved all procedures (Code 210/2015, date 3 March 2016). 

Dogs presenting with cough and diagnosed with TC were eligible for the study if an indication for the use of anti-inflammatory therapy was found. Tracheal collapse was diagnosed based on the objective verification of changes in tracheal diameter on radiographs, fluoroscopy and computed tomography, or the identification of tracheal collapse during endoscopy. When endoscopy could not be performed, the diagnosis was based on a combination of the following criteria: lack of clinical or radiographic evidence of an aspiration-related foreign body or neoplastic disease, and the observation of luminal changes in the tracheal diameter on radiographs or fluoroscopy. 

A 13-point grading system was established to quantify the severity of the clinical signs and to objectify the indication for the use of anti-inflammatory therapy. This system was based on four key clinical parameters (respiratory distress, cough episodes, cough frequency and tracheal sensitivity) whose variations were standardized according to a predetermined semiquantitative scale (Table 1). The sum of the scores obtained in each determined the final clinical score (FCS) at each control point at the hospital. According to this system, the researchers agreed and standardized that if the patient obtained more than 6 points for their FCS, they were considered a candidate to receive corticosteroids and entry into the study was offered. If the owner agreed, then the patient was randomized to the inhaled group (IG) or the oral group (OG). For this, an equal number of folded ballots with the inscription “oral group” and “inhaled group” were placed in an envelope. Once the invitation to enter the study was accepted, the owners were invited to draw one of these ballots and they were assigned to the group that their choice had randomly determined.

The treatment regimen for each group is specified in Table 2. For entry into the study, a pharmacological washout period of anti-inflammatory drugs of at least 15 days prior was established. The exclusion criteria for the patients were contraindication to the use of glucocorticoids (inhaled or oral), presence of comorbidities that could interfere with the results (particularly those that could induce polyuria–polydipsia and other clinical signs not directly attributable to tracheal collapse), lack of follow-up of the controls and interruption/variation of the therapeutic protocols. 

Regardless of the assigned therapeutic scheme, the following general treatment guidelines were applied to all dogs: avoid the use of collars in favor of harnesses, weight control and avoid exposure to tobacco smoke (or other environmental irritants) as well as sudden changes in behavior and environmental temperature. Dogs that were taking antitussives for more than one month prior to study entry were allowed to continue taking them as long as no contraindications (i.e., bronchiectasis) were identified during the diagnosis investigation. The use of additional medications was not allowed during the study period. The method of oral medication administration was free, and adhered to the previous habits of the owner in relation to their own animal. In the case of the inhalation therapy, owners were instructed to standardize their administration, following these guidelines:
-Use of a self-manufactured face mask made from plastic bottles, so that the lip corners are covered, the eyeballs are excluded, and 2–3 cm is maintained from the nose to the opening hole of the mask (Figure 1).-Pediatric or veterinary commercial valved holding chamber (VHC).-Once the metered-dose inhaler (MDI) dispenser, VHC and mask had been coupled, it was recommended to hold the animal minimally during administration. Once the indicated dose had been released, the system had to be kept coupled until 7–10 respiratory cycles had been completed.

The research team conducted a personal training session with the owners of each dog included in the study to teach them the correct method of use of the inhaled medication. The owners were instructed to progressively acclimate the dog to the facemask before activating the MDI.

After inclusion and random assignment to one study group (OG or IG), the animals were scheduled for face-to-face veterinary control on weeks 2 and 4 of treatment, while on weeks 1 to 4, the owners had to complete the home control form. The data collected in the veterinary controls were formalized by completing an inclusion form and another for follow-up (Table 1). Likewise, data collection for the home checkpoints carried out by the owners was completed using a specifically designed home control form (Table 1), which also included general information about the study and the treatment guidelines according to the assigned group.

A power calculation was performed before enrolment based on a 13-point clinical grading system. A conservative assumption determined that a standard deviation (SD) of 1.3 (10%) would capture the greatest changes on this scale. The null hypothesis was defined as no significant change on the 13-point scale between study groups, and the alternative hypothesis was that there would be a change of at least 1.8 points in the total score. By performing a bilateral t-test using a power calculator [24], a sample size of 14 dogs by group afforded an alpha risk of 0.05 and a beta risk of 0.05.

Individual scores for each clinical variable included in the grading system as well as the FCS constituted the semiquantitative variables analyzed in the study. Body weight and age were analyzed as quantitative variables. Quantitative data are presented as the mean ± standard deviation and categorical data are expressed as proportions and percentages. 

Differences between checkpoints for the scores of each clinical variable and FCS were explored by means of the Wilcoxon test for paired data. The internal percentage distribution of clinical variables between control points and between treatment groups was also analyzed by the comparison of proportions using a chi-squared test and column-paired analysis using Z-tests. 

Statistical analyses were performed using the commercially available statistical software package IBM SPSS Statistics 28.0.1.1 (14) and significance was set at *p* < 0.05.

## 3. Results

A total of 32 dogs were enrolled in the study. However, two dogs in the OG were excluded from the study due to a lack of owner compliance. Finally, the study included 30 dogs, 14 in the OG and 16 in the IG. The mean age of the 30 dogs was 7.0 ± 3.84 (1–16) years. Fourteen dogs were male (eight castrated) and sixteen were female (ten spayed). The weight range for all dogs was between 1.8 and 15 kg (mean, 4.72 kg), and the 5-point body condition score ranged between 2 and 5 (mean, 3.14). Represented breeds included Yorkshire terrier (9), mixed breed dogs (9), Maltese (3), Poodle (4), Chihuahua (3), and Pomeranian (2). 

At the time of enrolment, no significant differences between the two study groups in demographic variables or clinical parameters (respiratory distress, cough frequency, cough episodes, tracheal sensitivity and FCS) were found (Figure 2, Table 3).

### 3.1. Inhaled Group

Scores of each of the clinical variables monitored at the hospital (respiratory distress, cough episodes, cough frequency and tracheal sensitivity) as well as FCS were significantly reduced from the inclusion checkpoint compared with the week 2 and week 4 checkpoints (Table 3). Reductions in all these scores were still significant between week 2 and week 4. Paired column comparisons between the sample distributions of inner categories of each variable also showed this tendency. Thus, at inclusion, the percentage of dogs without episodes of respiratory distress (12.5%) was significantly lower than at the end of the study (week 4, 56.3%, *p <* 0.05). The cough episodes were paroxysms in 93.8% of dogs at inclusion, and this percentage was significantly lower at weeks 2 (31.3%, *p <* 0.01) and 4 (6.3%, *p <* 0.01). Cough frequency was also reduced from persistent and very frequent to frequent and sporadic. It is striking that none of the dogs in the group presented cough sporadically at inclusion and week 2, while at week 4, 75% presented this cough frequency category. The differences in the inner distributions of tracheal sensitivity categories followed a similar trend, although they became more evident and significant when observing the difference in the percentage of dogs with hyperreactive tracheas at the time of inclusion (43.8%) compared to weeks 2 (6.3%) and 4 (0%). At inclusion, 75% of dogs were in the severe score class, but at the end of the study, 100% of them were in the mild score class. 

Water consumption and urinary frequency did not significantly change during the study in this group (Figure 3). The adherence to inhaled treatment was also stable during the study, with a majority (87.5%) of dogs always accepting the dosage prescribed. The tolerance was always good or excellent, but a progressive and sustained improvement was observed from week 1 (68.8% excellent, 31.3% good) to week 4 (87.5% excellent, 12.5% good).

### 3.2. Oral Group 

The group of dogs that was randomized to receive oral corticosteroid therapy showed overall favorable results equivalent to those described for the IG (Table 3). The scores of each of the clinical variables controlled at the hospital (respiratory distress, cough episodes, cough frequency and tracheal sensitivity) as well as FCS were significantly reduced from those at the inclusion point at the 2- and 4-week checkpoints (Table 3). All except the type of cough episodes were also significantly reduced between week 2 and week 4. Paired column comparisons between the sample distributions of the inner categories of each variable implied the same overall state as in the inhaled group. That is, a significant reduction in the percentages of appearance in the categories associated with a greater severity of clinical signs in favor of those of less severity from the point of inclusion, through the control of week 2 and up to that of week 4, was found. The specific percentages and statistical results of the paired comparison are shown in Table 3.

At week 1, the perceptions of the owners of 35.7% of the dogs from this group were that water consumption and urinary frequency increased compared to the inclusion point (Figure 3). This percentage was reduced at week 2 (28.5%) but again increased at weeks 3 (42.9% for water consumption, 50% for urinary frequency) and 4 (50%). These changes were not significant according to between-control point statistical comparisons. 

Adherence to the treatment was excellent, with 92.9% of owners declaring that they were always able to administer the prescribed dosages during the study. Tolerance was also excellent, with mild differences during the study (85.7% reported excellent at weeks 1 and 2, and 92.9% excellent at weeks 3 and 4).

### 3.3. Comparison between Groups (Inhaled–Oral)

As previously mentioned, significant improvements in the clinical study parameters were identified in both groups throughout the study. However, between-group (inhaled–oral) comparisons at each checkpoint revealed no statistically significant differences, thus indicating that the improvements provided by both regimens tested for the treatment of cough in these groups of dogs with TC were equivalent. However, the only significant difference between the groups was that the FCS in the IG was lower (5.69 ± 0.79) than in the OG (6.43 ± 1.02, *p* < 0.05; Table 3) at the study’s endpoint.

The adherence and tolerance to both treatment schemes were also equivalent, without significant differences between groups at any check point. However, from week 2 until the end of the study, dogs in the OG were significantly thirstier and urinated more frequently than dogs in the IG (Figure 3). 

## 4. Discussion

This study has prospectively compared the effects of oral and inhaled corticosteroids in a group of dogs with cough and TC. The results provide evidence that encourages the use of inhaled corticosteroids, since equivalent efficacy was evidenced with excellent tolerance and adherence to the inhaled method and without typical secondary signs of cortisol excess.

Recently, a prospective placebo-controlled cross-over study in dogs with inflammatory airway disease demonstrated the efficacy of inhaled fluticasone in the clinical control of chronic cough [20]. Cough duration, frequency, and severity were significantly decreased at the end of the study. The feasibility of aerosolized delivery improved with continued use. These results are consistent with those obtained in the group of dogs that received inhaled fluticasone in the present study. However, there were some notable differences in the therapeutic designs of both studies. The duration of the study and the treatment scheme used in our study were different, since a higher dose (100 µg/8 h) was initially given, which was then gradually reduced until the end of the study (4 weeks), while in the study by Chan and Johnson [20], a stable regimen of 110 μg/12 h was used for 6 weeks. It is possible that higher doses were considered in that study since one of the inclusion criteria was that the dogs had had a cough for at least two months. Previous studies carried out in cats with experimental asthma compared various fluticasone dosages (44, 110 and 220 μg/12 h) without finding significant differences in the inflammation parameters evaluated in bronchoalveolar lavage [14]. Although there is no clearly established consensus, a dose of around 110 μg/12 h has previously been estimated to be appropriate in dogs [25,26]. Since it has also been established that a certain amount of time is required for owners and animals to get used to the inhaled treatment protocol [20,25], in the present study, a higher administration frequency was designed at the beginning (every 8 h). Beyond the differences in the specific design of the therapeutic plan, the evidence on the clinical improvement of cough in patients with airway inflammation found in our study is comparable to that of previous studies. This suggests that, in a clinical setting, it is appropriate to adapt the dose, frequency and duration of the fluticasone treatment to the chronicity of the cough and the presumed feasibility of the administration of the inhaled medication in each specific patient.

Fluticasone has high local deposition [23] and potency in the airways, and theoretically reduced systemic adverse effects because of decreased absorption and increased rate of first-pass metabolism [22]. A recent study on coughing in dogs caused by airway inflammation documented some hypercortisolism-related adverse effects of fluticasone treatment in 50% of cases administered for 6 weeks at a dose of 110 μg/12 h [20]. In dogs with idiopathic eosinophilic bronchopneumopathy treated with inhaled fluticasone (100–250 μg/12 h), long-term follow-up found HPAA inhibition in two dogs treated for more than 2 years with fluticasone monotherapy; only one dog had clinical signs of iatrogenic hyperadrenocorticism [26]. Other possible local side effects have been anecdotally described in dogs undergoing chronic treatment with inhaled fluticasone [27]. In the present study, a short-term-oriented dosing scheme was used. Although initially starting with doses even higher than those used in other studies (100 µg/8 h), these doses were soon reduced at 5-day intervals, providing adequate control of cough without the typical signs of iatrogenic hyperadrenocorticism. In contrast, although the doses of prednisone used were also low and with an early reduction scheme, these signs did occur (according to the perception of the owners) with a significantly higher frequency in the group that received oral prednisone. A previous retrospective study documented owner-reported signs of hypercortisolism in five dogs treated orally with prednisolone, but the same owners did not note any adverse effects in the same dogs when they were administered fluticasone [28]. In healthy dogs treated with inhaled fluticasone, the asymptomatic inhibition of HPAA has been shown to occur after 3 to 4 weeks of treatment [22,29]. Therefore, inhaled fluticasone therapy using a conservative dosing schedule as used in this study may provide short-term control of cough in dogs with TC without inducing clinically apparent side effects, although the subclinical inhibition of the HPAA cannot be ruled out.

Although histopathological studies on dogs with TC were carried out more than 30 years ago [5,6], there is a consensus that it is a degenerative disease that affects the tracheal cartilage, for which there is no inflammatory basis [2,7,8]. There is also growing evidence that it constitutes a main component of a clinical syndrome in which other comorbidities (other types of airway collapse, chronic bronchitis, bronchiectasis, myxomatous mitral valve disease, congestive heart failure and pulmonary hypertension) usually concur to determine the appearance of the final clinical signs [7,30,31]. The adequate identification and therapeutic control of these comorbidities is a fundamental aspect of the standard of care for dogs with TC. However, the presence of cough is one of the most frequent clinical signs, and one that has the greatest influence on the quality of life of dogs with TC in their home environment. A recent study employed surveys to investigate the usual practices of veterinary specialists (American College of Veterinary Internal Medicine and Cardiology diplomates, European College of Veterinary Internal Medicine—Companion Animal diplomates, and American College of Veterinary Emergency and Critical Care diplomates) in the diagnosis and treatment of dogs with TC [4]. Almost 100% of respondents (n = 180) recognized that they used glucocorticoids for stable, coughing dogs diagnosed with TC over the previous year. Of those prescribing glucocorticoids, 78% used prednisone or prednisolone (most frequently), and only 21% used inhaled fluticasone as a primary option. The most common prednisone or prednisolone starting dose was 1 mg/kg/24 h (69% of respondents), although 21% began treatment at 0.5 mg/kg/24 h, like those used as starting doses in the present study. Respondents were divided over the anticipated length and duration of the treatment, as follows: 50% foresaw that dogs would have to be continued indefinitely on a low maintenance dose, and 49% used a tapering dose with the goal of discontinuing glucocorticoid treatment [4]. Although there are no similar data derived from non-specialist veterinarians, it is expected that between them, the rate of use of oral glucocorticoids will be similar or even higher. The results of the present study show that treatment with inhaled fluticasone in dogs with TC has achieved parameters of clinical improvement comparable to oral corticosteroids. Although we did not include cytomicrobiological studies of the airways (which would have been the ideal way to establish the presence and type of inflammation present), the most favorable clinical response to anti-inflammatory therapy in both groups suggests that the presence of associated airway inflammation constitutes a key determinant in the appearance and severity of typical TC clinical signs. Modern studies are required to re-evaluate the importance of chronic airway inflammation in dogs with TC. In the meantime, the findings of the present study encourage the use of the inhaled versus oral route in dogs with cough and TC.

The present study has several limitations. Fluticasone and prednisone therapies were assessed for only 4 weeks, and longer-term follow-ups are required, since relapses may occur and the appearance of signs of hyperadrenocorticism must be evaluated during continuous treatments that take place over long periods of time. Furthermore, we relied on owner compliance and perception when inferring adverse effects, feasibility and response to treatment; however, owner perception is also a cornerstone when re-evaluating dogs with a chronic cough in the clinic. Of note, additional medications were permitted at the discretion of the clinician with consideration of ethical concerns. Not all dogs underwent the same type of complementary examinations, and we cannot entirely exclude the existence of underlying infections or other causes of cough other than TC. In particular, it would have been desirable to systematically obtain samples of the airways for cytomicrobiological studies in order to specify and characterize the presence of inflammation and the therapeutic response.

## 5. Conclusions

The administration of two protocols of therapy, inhaled fluticasone and oral prednisone, to two groups of dogs with cough and TC as well as similar clinical and demographic characteristics provided comparable efficacy with less incidence of signs of hypercortisolism (increased thirst and urination), and very good tolerance to the inhaled administration protocol. These data support and encourage using inhaled fluticasone to treat cough in dogs with TC.

## Figures and Tables

**Figure 1 vetsci-10-00548-f001:**
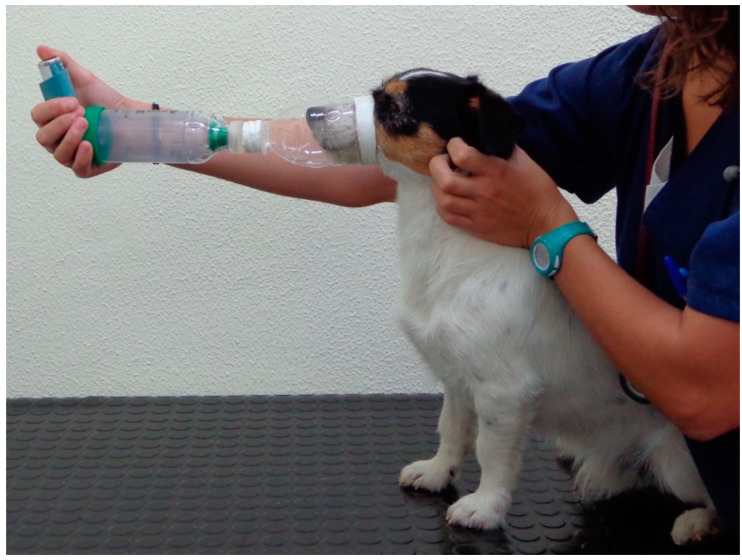
Example of device configuration (valved holding chamber and face mask) and patient restraint protocol used for the administration of inhaled therapy in the study.

**Figure 2 vetsci-10-00548-f002:**
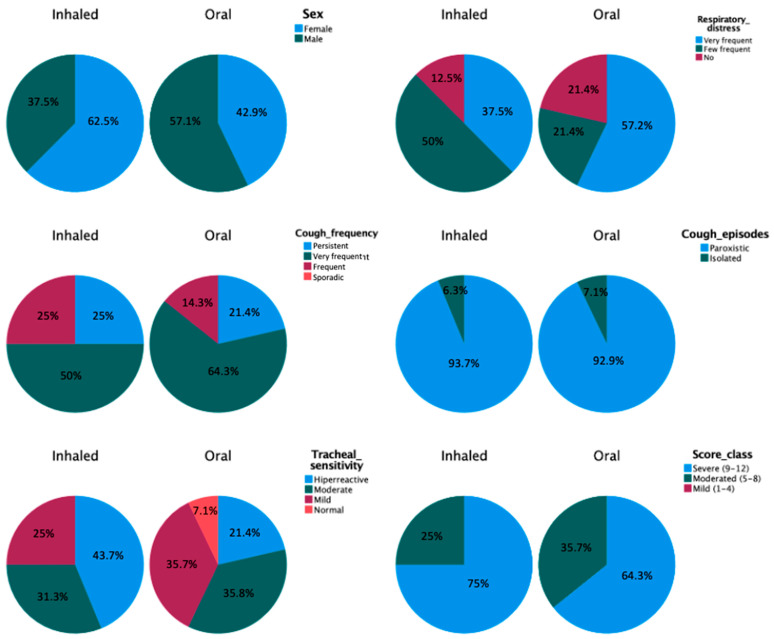
Comparisons of distribution of sex and clinical parameters between study groups at the time of inclusion.

**Figure 3 vetsci-10-00548-f003:**
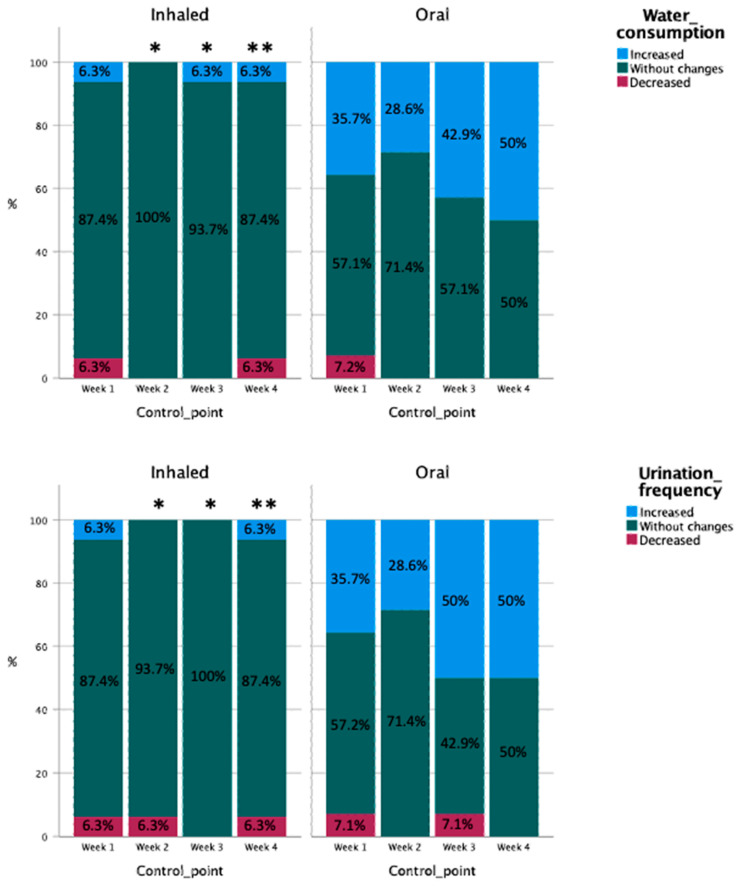
Comparisons of distribution percentages between categories of water consumption between the different control points in both study groups. Asterisks in the top of a column indicate differences from the same control point of the other therapy group, one if *p* < 0.05 and two if *p* < 0.01.

**Table 1 vetsci-10-00548-t001:** The grading system used in the study.

Control Points	Clinical Parameter	Score	Options	Description
Inclusion, Weeks 2 and 4 (veterinary form)	Respiratory distress			
		1	No	Absence
		2	Few frequent	<5 episodes/week
		3	Very frequent	≥5 episodes/week
	Cough episodes			
		1	Isolated	Unlinked coughing fits
		2	Paroxysmic	Linked coughing fits
	Cough frequency			
		1	Sporadic	<5 episodes/week
		2	Frequent	≥5 episodes/week
		3	Very frequent	≥1 episode/day
		4	Persistent	Coughing most of the day
	Tracheal sensitivity to soft tracheal compression			
		1	Normal	No cough response
		2	Mild	<3 isolated coughing fits
		3	Moderate	≥3 isolated coughing fits
		4	Hyperreactive	Coughing paroxysm
	Total score	4–13	≥6: Inclusion invitation	
	Score class	<6	Mild	
		6–9	Moderate	
		>9	Severe	
Weeks 1 to 4 (owner form)	Water consumption			
		+1	Increased	More than usual
		−1	Decreased	Less than usual
		0	Without changes	As usual
	Urinary frequency/volume			
		+1	Increased	More than usual
		−1	Decreased	Less than usual
		0	Without changes	As usual
	Adherence			
		1	Never	Grading scale
		2	Sometimes	
		3	Usually	
		4	Always	
	Tolerance			
		1	Bad	Grading scale
		2	Middle	
		3	Good	
		4	Excellent	

**Table 2 vetsci-10-00548-t002:** Therapeutic regimens of each study group.

Study Group	Product	Dose	Frequency	Duration
Inhaled	Fluticasone	100 μg/dog	8 h	5 days
		100 µg/dog	12 h	5 days
		100 µg/dog	24 h	5 days
		100 µg/dog	48 h	5 days
		50 µg/dog	48 h	10 days
Oral	Prednisone	0.5 mg/kg	12 h	3 days
		0.25 mg/kg	12 h	5 days
		0.25 mg/kg	24 h	10 days
		0.25 mg/kg	48 h	12 days

**Table 3 vetsci-10-00548-t003:** Descriptive statistics (see description below) and results of pair-to-pair comparisons for the clinical parameters between control points in both study groups.

	Inhaled			Oral		
	Inclusion	Week 2	Week 4	Inclusion	Week 2	Week 4
	A	B	C	D	E	F
**Respiratory distress**	2.25 ± 0.68 **B,C**	1.69 ± 0.48 C	1.44 ± 0.51	2.36 ± 0.84 E,F	2.07 ± 0.73 F	1.79 ± 0.58
No episodes	2/16 (12.5)	5/16 (31.3)	9/16 (56.3) A	3/14 (21.4)	3/14 (21.4)	4/14 (28.6)
Few frequent	8/16 (50)	11/16 (68.8)	7/16 (43.8)	3/14 (21.4)	7/14 (50)	9/14 (64.3)
Very frequent	6/16 (37.5)	0	0	8/14 (61.5) F	4/14 (28.6)	1/14 (7.1)
**Cough** **episodes**	1.94 ± 0.25 **B,C**	1.31 ± 0.48 C	1.06 ± 0.25	1.93 ± 0.27 **E,F**	1.29 ± 0.47	1.07 ± 0.27
Isolated	1/16 (6.3)	11/16 (68.8) **A**	15/16 (93.8) **A**	1/14 (7.1)	10/14 (71.4) **D**	13/14 (92.9) **D**
Paroxysmic	15/16 (93.8) **B,C**	5/16 (31.3)	1/16 (6.3)	13/14 (92.9) **E,F**	4/14 (28.6)	1/14 (7.1)
**Cough frequency**	3.0 ± 0.73 **B,C**	2.25 ± 0.45 **C**	1.31 ± 0.6	3.1 ± 0.62 **E,F**	2.29 ± 0.47 F	1.71 ± 0.47
Sporadic	0	0	12/16 (75)	0	0	4/14 (28.6)
Frequent	4/16 (25)	12/16 (75) A,**C**	3/16 (18.8)	2/14 (5.4)	10/14 (71.4) D	10/14 (71.4) D
Very frequent	8/16 (50) C	4/16 (25)	1/16 (6.3)	9/14 (52.9)	4/14 (28.6)	0
Persistent	4/16 (25)	0	0	3/14 (21.4)	0	0
**Tracheal sensitivity**	3.19 ± 0.83 **B,C**	2.44 ± 0.63 C	1.87 ± 0.62	2.71 ± 0.91 E,**F**	2.21 ± 0.58 F	1.86 ± 0.53
Normal	0	0	4/16 (25)	1/14 (7.1)	1/14 (7.1)	3/14 (21.4)
Mild	4/16 (25)	10/16 (62.5)	10/16 (62.5)	5/14 (35.7)	9/14 (64.3)	10/14 (71.4)
Moderate	5/16 (31.3)	5/16 (31.3)	2/16 (12.5)	5/14 (35.7)	4/14 (28.6)	1/14 (7.1)
Hyperreactive	7/16 (43.8) B	1/16 (6.3)	0	3/14 (21.4)	0	0
**Total score**	10.38 ± 1.63 **B,C**	7.69 ± 1.01 **C**	5.69 ± 0.79	10.07 ± 1.39 **E,F**	7.86 ± 1.46 **F**	6.43 ± 1.02 C
**Score class**						
Mild	0	1/16 (6.3)	16/16 (100)	0	3/14 (21.4)	8/14 (57.1)
Moderate	4/16 (25)	15/16 (93.8) **A**	0	5/14 (35.7)	9/14 (64.3)	6/14 (42.9)
Severe	12/16 (75)	0	0	9/14 (64.3) **E**	2/14 (14.3)	0

NOTE: Capital letters under a data box indicate significant differences (*p* < 0.05) between columns based on two-tailed tests (bold if *p* < 0.01). For each significance pair, the key for the category with the smallest proportion in the column appears in the category with the largest proportion. In cells with a percentage of zero, differences were assumed to be obvious, and these cells were not used in two-way comparisons by column. For each variable, the first row corresponds to the mean value and standard deviation of the score, and the following rows correspond to the ratios by number of cases and percentages (in brackets) of each category.

## Data Availability

Not applicable.

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
