# Peer review of "Comparative Study of Inhaled Fluticasone Versus Oral Prednisone in 30 Dogs with Cough and Tracheal Collapse"

_vetsci, 2023, doi:10.3390/vetsci10090548_

Round 1
Reviewer 1 Report
Dear authors,
Here my contribution.
Abstract
Line 30 (and Line 192, 218, 219,.....)- statistical p in italics
Keywords: Please explain MDI significance
please add: Fluticasone propionate and Prednisone
Introduction
What is the citation order of the references, chronological or alphabetical?
Line 40 - (Clercx, 2017; Scansen and Weisse, 2014) but line 45 - (Macready et al, 2007; Carr et al, 2023) or Line 48 - (Dallman et al, 1985; Dallman et al, 1988; Mason and Johnson, 2004; Clercx, 2017; Macready et al, 2007; Johnson and Pollard, 2010).
From Veterinary Sciences instructions for authors (https://www.mdpi.com/journal/vetsci/instructions)
- References: References must be numbered in order of appearance in the text (including table captions and figure legends) and listed individually at the end of the manuscript. We recommend preparing the references with a bibliography software package, such as EndNote, ReferenceManager or Zotero to avoid typing mistakes and duplicated references. We encourage citations to data, computer code and other citable research material. If available online, you may use reference style 9. below.
Lines 52 to 54 - Structurally, the tracheal rings of affected dogs show deficiencies of glycoproteins, glycosaminoglycans, and calcium and decreased water content, resulting in loss of cartilage rigidity.
Line 52 - cardiopulmonary diseases but Line 65 - cardiorespiratory conditions ????
Line 82 - sarcopenia or miopenia instead muscle atrophy,....
See Myopenia—a new universal term for muscle wasting - J Cachexia Sarcopenia Muscle. 2011 Mar; 2(1): 1–3. doi: 10.1007/s13539-011-0025-7
Materials and Methods
Line 133 - Table 1 should only be presented in the text after all the information contained therein has appeared in the text and not as it occurs in this version of the manuscript.
Results
Figure 4 - asterisks no see
Despite already having a good level of scientific English, the text must be proofread by a native speaker.
I leave you an example:
Lines 52 to 54 - Structurally, the tracheal rings of affected dogs show deficiencies of glycoproteins, glycosaminoglycans, and calcium and decreased water content, resulting in loss of cartilage rigidity.
Author Response
RESPONSE TO REVIEWER 1
Dear authors,
Here my contribution.
Response: We thank the reviewer for his perspective and assessment of our study. His suggestions will undoubtedly contribute to improving the final version that we hope it can be published.
Abstract
Line 30 (and Line 192, 218, 219,.....)- statistical p in italics
Response: corrected according to the reviewer's suggestion
Keywords: Please explain MDI significance
please add: Fluticasone propionate and Prednisone
Response: corrected according to the reviewer's suggestion.
Introduction
What is the citation order of the references, chronological or alphabetical?
Line 40 - (Clercx, 2017; Scansen and Weisse, 2014) but line 45 - (Macready et al, 2007; Carr et al, 2023) or Line 48 - (Dallman et al, 1985; Dallman et al, 1988; Mason and Johnson, 2004; Clercx, 2017; Macready et al, 2007; Johnson and Pollard, 2010).
From Veterinary Sciences instructions for authors (https://www.mdpi.com/journal/vetsci/instructions)
References: References must be numbered in order of appearance in the text (including table captions and figure legends) and listed individually at the end of the manuscript. We recommend preparing the references with a bibliography software package, such as EndNote, ReferenceManager or Zotero to avoid typing mistakes and duplicated references. We encourage citations to data, computer code and other citable research material. If available online, you may use reference style 9. below.
Response: We apologize for this unfortunate formal deficiency. The references have now been introduced numbered in order of appearance in the text.
Lines 52 to 54 - Structurally, the tracheal rings of affected dogs show deficiencies of glycoproteins, glycosaminoglycans, and calcium and decreased water content, resulting in loss of cartilage rigidity.
Response: Corrected. The sentence has been rewritten as suggested by the reviewer.
Line 52 - cardiopulmonary diseases but Line 65 - cardiorespiratory conditions ????
Response: Corrected. The allusion to this group of diseases has been homogenized as "cardiorespiratory diseases"
Line 82 - sarcopenia or miopenia instead muscle atrophy,....
See Myopenia—a new universal term for muscle wasting - J Cachexia Sarcopenia Muscle. 2011 Mar; 2(1): 1–3. doi: 10.1007/s13539-011-0025-7
Response: Corrected. The term 'myopenia' has been used instead of 'muscular atrophy' as suggested by the reviewer. We appreciate this clarification that we will also use in the future.
Materials and Methods
Line 133 - Table 1 should only be presented in the text after all the information contained therein has appeared in the text and not as it occurs in this version of the manuscript.
Response: The situation of Table 1 in the version evaluated by the reviewer corresponds to a version generated (probably automatically) by the journal system itself, from the Word file that we sent, in which the tables and figures were at the end of the text. We believe that the table has been automatically introduced after the first allusion to it in the text. It is not unusual that due to requirements of adaptation to the editorial format, a table or a figure is placed like this and references to them continue to appear later in the text. Anyway, we have added a comment to ask the editor that table 1 be placed as the reviewer suggests if this would be possible.
Results
Figure 4 - asterisks no see
Response: This figure does not include asterisks because for adherence and tolerance no significant differences were found between therapies at any control point. Anyway, this figure will be removed in the corrected version to attend to the suggestion of another reviewer who has considered that since there are no differences, it is enough to include the information in this regard in the text.
Comments on the Quality of English Language
Despite already having a good level of scientific English, the text must be proofread by a native speaker.
I leave you an example:
Lines 52 to 54 - Structurally, the tracheal rings of affected dogs show deficiencies of glycoproteins, glycosaminoglycans, and calcium and decreased water content, resulting in loss of cartilage rigidity.
Response: We have contracted the English editing service offered by the journal to address this criticism in which both reviewers have agreed. The revised version included the changes recommended by the English editing reviewer.

Reviewer 2 Report
vetsci-2529474-peer-review-v1
Review of: “Comparative study of inhaled fluticasone versus oral prednisone in 30 dogs with cough and tracheal collapse”.
The authors of this manuscript compared oral prednisone to aerosolized and inhaled fluticasone for the control of coughing in dogs with tracheal collapse.
The hypothesis and findings are interesting, helpful, and clinically relevant. The manuscript is worthy of publication (with changes) and the journal is appropriate.
The data supports the conclusions of the paper.
It was heartening to find both a power calculation and a hypothesis included in the manuscript.
My criticisms of the manuscript are as follows:
- Perhaps the biggest shortcoming is the fact that, although there was a demonstrated clinical effect on the coughing in this cohort, the assumption was made that the coughing was only due to collapsed trachea. We are not sure these dogs did not also have inflammatory lower airway disease (which could be common in a cohort like this).
- It is also possible that although the clinical sign of coughing was suppressed, if there was inflammatory airway disease if the inflammation is not dealt with, the airway pathology may be progressive (in the face of clinical sign suppression). This shortcoming should be acknowledged in the weaknesses paragraph of the discussion. The consequences of not including cytological evaluation of the airways and the response of the cytology to the treatment interventions in the investigations of this cohort would be an important aspect for the authors to discuss.
- The English language is poor and needs substantial editorial attention. The word ‘insidious’, meaning a slow and subtle progression of signs, is in my view not what the authors intend. Sentence structure and other word choices are also poor and need the attention of an English editor (not the role of a reviewer).
- In my view, Fig. 4 does not present anything significant and the findings can be dealt with in the text.
- Table 3 needs an explanation for the capital letters included among the numbers. Do these denote significance?
There is a need for extensive attention to the written language.
Author Response
RESPONSE TO REVIEWER 2
The authors of this manuscript compared oral prednisone to aerosolized and inhaled fluticasone for the control of coughing in dogs with tracheal collapse.
The hypothesis and findings are interesting, helpful, and clinically relevant. The manuscript is worthy of publication (with changes) and the journal is appropriate.
The data supports the conclusions of the paper.
It was heartening to find both a power calculation and a hypothesis included in the manuscript.
Response: We thank the reviewer for his perspective and assessment of our study. His suggestions will undoubtedly contribute to improving the final version that we hope can be published.
My criticisms of the manuscript are as follows:
Perhaps the biggest shortcoming is the fact that, although there was a demonstrated clinical effect on the coughing in this cohort, the assumption was made that the coughing was only due to collapsed trachea. We are not sure these dogs did not also have inflammatory lower airway disease (which could be common in a cohort like this).
It is also possible that although the clinical sign of coughing was suppressed, if there was inflammatory airway disease if the inflammation is not dealt with, the airway pathology may be progressive (in the face of clinical sign suppression). This shortcoming should be acknowledged in the weaknesses paragraph of the discussion. The consequences of not including cytological evaluation of the airways and the response of the cytology to the treatment interventions in the investigations of this cohort would be an important aspect for the authors to discuss.
Response: We agree with the reviewer that the cytomicrobiological evaluation of the airways would have been very useful complementary information in our study to establish the degree and type of airway inflammation. We have added a specific allusion in the discussion (lines 379-383) and in the limitations paragraph (lines 397-399).
The English language is poor and needs substantial editorial attention. The word ‘insidious’, meaning a slow and subtle progression of signs, is in my view not what the authors intend. Sentence structure and other word choices are also poor and need the attention of an English editor (not the role of a reviewer).
Response: The term “insidious” has been replaced for “persistent”. We have contracted the English editing service offered by the journal to address this criticism in which both reviewers have agreed. The revised version included the changes recommended by the English editing reviewer.
In my view, Fig. 4 does not present anything significant and the findings can be dealt with in the text.
Response: Figure 4 has been removed in the corrected version as suggested by the reviewer.
Table 3 needs an explanation for the capital letters included among the numbers. Do these denote significance?
Response: In the word file sent for review to the journal, Table 3 included a table footer explaining the meaning of the capital letters. The version evaluated by the reviewer corresponds to a version generated (probably automatically) by the journal system itself, from the Word file that we sent. This automatic conversion process probably mistakenly removed that footer which has now been reintroduced.
Comments on the Quality of English Language
There is a need for extensive attention to the written language.
Response: As we indicated above, we have contracted the English editing service offered by the journal to address this criticism in which both reviewers have agreed. The revised version included the changes recommended by the English editing reviewer.
